# Facilitators and barriers of physical activity, sedentariness and exercise adoption among healthcare professionals in Lagos State, Nigeria – A qualitative review

**Blossom Adaeze Maduafokwa**[1]*, **Mobolanle Balogun**[2],
**Kamaldeen Sunkanmi Abdulraheem**[1]

**1** Department of Community Health, Lagos University Teaching Hospital, Lagos, Nigeria, **2** Department of Community Health and Primary Care, University of Lagos, Lagos, Nigeria

\* bmaduafokwa@yahoo.com

## Abstract

Physical inactivity is a major global public health concern, identified by the World Health Organization as the fourth leading cause of mortality worldwide and associated with an estimated 3.2 million deaths annually. Healthcare professionals (HCPs) serve as frontline advocates for health promotion and disease prevention, yet many struggle to maintain adequate levels of physical activity themselves, which not only compromises their health but also diminishes their capacity to counsel patients effectively. This qualitative study explored the facilitators and barriers of physical inactivity, sedentariness, and exercise adoption among healthcare professionals in Lagos State, Nigeria, using the Social Ecological Model (SEM) to examine influences at individual, interpersonal, organizational, community, and policy levels. Data were collected through six focus group discussions with healthcare professionals, four in-depth interviews with family members of HCPs, and five key informant interviews with healthcare facility heads and policy makers in urban planning in Lagos State. Findings revealed that time constraints, heavy workloads, inadequate infrastructure, safety concerns, and cultural norms were major barriers to physical activity, while facilitators included peer support, leadership engagement, self-efficacy, and incentives such as free gym access. Organizational strategies such as provision of onsite gyms at healthcare facilities, scheduled breaks, and promotion of a workplace fitness culture were identified as critical enablers. The study highlights the urgent need for comprehensive, multi-level interventions to address barriers and strengthen facilitators of physical activity among healthcare professionals. Addressing these gaps is essential not only for improving their health but also for enhancing their role as credible advocates of active lifestyles.

**Data availability statement:** All relevant data are within the paper and its Supporting Information files.

**Funding:** The author(s) received no specific funding for this work.

**Competing interests:** The authors have declared that no competing interests exist.

## Introduction

Non-communicable diseases (NCDs) remain the leading causes of mortality worldwide, accounting for seven of the ten top causes of death globally. [1] The World Health Organization (WHO) estimates that physical inactivity, a major modifiable risk factor for NCDs, accounts for more than 5 million deaths annually, making it one of the leading modifiable risk factors for noncommunicable diseases worldwide. [2] Physical inactivity has been described as a global "pandemic" with profound health, social, and economic consequences. [3] Sedentariness is now recognized as a major global public health problem, contributing substantially to morbidity and mortality. Prolonged sitting and physical inactivity are strongly associated with increased risk of cardiovascular disease, type 2 diabetes, obesity, metabolic syndrome, and certain cancers, independent of overall physical activity level. [4] While quantitative studies have consistently linked physical inactivity and sedentariness to cardiovascular disease, diabetes, obesity, cancers, and premature mortality, these findings alone do not explain why individuals remain inactive or what prevents them from adopting more active lifestyles [5]. A deeper, qualitative understanding of the barriers and facilitators of physical activity is needed to design interventions that are context-specific, sustainable, and effective. In low- and middle-income countries such as Nigeria, rapid urbanization, constrained built environments, and work-related factors exacerbate sedentary behaviors, further amplifying the burden of preventable chronic diseases. [6]

Healthcare professionals (HCPs) occupy a unique position in this discourse. As health advocates and role models, they influence patient behaviors and public health outcomes. However, many HCPs themselves face challenges in maintaining adequate levels of physical activity due to long working hours, high patient loads, inadequate facilities, and competing responsibilities. [7] Globally, qualitative studies have revealed recurring themes such as lack of time, motivational barriers, workplace cultures, and infrastructural limitations as impediments to physical activity. [8] Facilitators often include peer support, leadership involvement, self-efficacy, and access to supportive environments. Yet, in Nigeria, little qualitative research has explored how these factors specifically shape the experiences of HCPs. [8]

This gap is particularly concerning given Nigeria's rising burden of NCDs and the critical role of HCPs in health promotion. Exploring their lived experiences through qualitative inquiry offers nuanced insights into the structural, cultural, and interpersonal factors that shape their ability to engage in physical activity. Such evidence is vital not only for improving the health of HCPs themselves but also for enhancing their effectiveness as credible advocates of exercise adoption among patients and communities.

This study therefore examines the facilitators and barriers of physical activity, sedentariness and exercise adoption among healthcare professionals in Lagos State, Nigeria using the Social Ecological Model (SEM) as a guiding framework. It highlights the multiple levels of influence – individual, interpersonal, organizational, community, and policy – that either hinder or promote physical activity, sedentariness and exercise adoption. By centering the voices of healthcare professionals, this qualitative review contributes to the growing body of literature on physical activity promotion while generating context-specific insights that can inform workplace wellness strategies and health policy in Nigeria.

## Conceptual framework

This study applies the Social Ecological Model (SEM) to explore facilitators and barriers of physical activity, sedentariness and exercise adoption among healthcare professionals in Lagos State. The SEM provides a multi-level framework, recognizing that physical activity is shaped by interactions between individuals and their social, organizational, community, and policy environments. [9] At the individual level, knowledge, attitudes, self-efficacy, and health status influence behavior, while interpersonal factors such as peer encouragement and family support can either promote or hinder participation. [9] Organizational factors, including workload, workplace culture, and availability of workout facilities, affect opportunities for exercise, while community-level determinants such as infrastructure, safety, and cultural norms further shape engagement. [9] At the policy level, institutional and governmental actions – such as workplace wellness programs or investment in recreational spaces – create enabling or restrictive environments. [9] The SEM emphasizes that effective and sustainable adoption of physical activity requires interventions that act simultaneously across these levels. [9] Its application in this study provides a holistic understanding of the contextual factors influencing physical activity, sedentary and exercise behavior among healthcare professionals and highlights priority areas for targeted intervention.

## Materials and methods

### Study design

This study employed a qualitative, exploratory design using focus group discussions (FGDs), in-depth interviews (IDIs), and key informant interviews (KIIs).

### Setting

This study was conducted in Lagos State, Nigeria, the country's most populous state and commercial hub. Created in 1967, Lagos lies in the southwest region and borders Ogun State to the north and east, the Republic of Benin to the west, and the Atlantic Ocean to the south. With an estimated population of 35 million by 2020, Lagos is a socio-cultural and economic melting pot that attracts residents from across Nigeria and beyond. Administratively, Lagos consists of 20 Local Government Areas (LGAs), of which 16 are classified as urban and four as rural. Healthcare in the state is provided through a three-tier system comprising 177 primary facilities, 29 secondary facilities, and 11 tertiary institutions, which include teaching hospitals, general hospitals, and specialist centers. These facilities employ a wide range of healthcare professionals (HCPs), including doctors, nurses, pharmacists, physiotherapists, laboratory scientists, community health officers, and allied health staff.

### Study population

The study population comprised diverse stakeholders across multiple levels of the Social Ecological Model. Participants included healthcare professionals (such as doctors, nurses, and allied health workers) representing the individual level; family members of healthcare professionals reflecting the interpersonal and household influences; healthcare facility leaders and administrators representing the organizational level; and policymakers as well as urban planning officials who operate at the institutional and policy levels. Participant recruitment for this study took place from March 30, 2024, to July 14, 2024.This broad inclusion ensured a comprehensive understanding of the facilitators and barriers to physical activity, sedentariness, and exercise adoption among healthcare professionals within their personal, professional, and structural environments

### Inclusion and exclusion criteria

For focus group discussions (FGDs), HCPs were eligible if they had been employed in public facilities for at least six months, were aged 18 years and above, and were willing to participate. Pregnant women, severely ill HCPs, or those

unable to contribute meaningfully were excluded. In-depth interviews (IDIs) included family members or close friends of HCPs who were familiar with their lifestyle and work schedules, aged 18 years and above, and resident in Lagos. Key informant interviews (KIIs) targeted healthcare leaders and policymakers with at least two years of relevant experience and knowledge of workplace or public health policies affecting physical activity.

## Sampling and sample size

Purposive sampling was used to recruit participants with relevant experiences. Participant recruitment took place between March 30 and July 14, 2024. Multiple recruitment strategies were employed. For HCPs, facility gatekeepers (medical directors, heads of departments, and nursing unit heads) facilitated initial access to eligible participants within public healthcare facilities. Following approval, eligible HCPs were approached through direct invitations during staff meetings and via departmental WhatsApp groups. Interested participants contacted the research team directly or were referred by gatekeepers. Family members were recruited through participating HCPs, who were asked to identify adult relatives or close associates familiar with their daily routines and lifestyle behaviors. Healthcare leaders and policymakers were recruited through direct email and telephone invitations based on their roles and involvement in workplace health promotion, public health, or urban planning. Participation was voluntary, and written informed consent was obtained prior to data collection. For the focus group discussions (FGDs), six groups comprising 6–9 participants each were conducted (total n = 39). Groups were stratified by professional cadre (two groups each for doctors, nurses, and community health officers/extension workers) and further categorized into exercise adopters and non-adopters to facilitate comparative analysis. Approximately 58 HCPs were approached, of whom 39 consented and participated. Non-participation was primarily due to time constraints, work schedule conflicts, or inability to attend scheduled sessions.

In-depth interviews (IDIs) were conducted with four family members of HCPs. Six individuals were invited, with two declining participation due to scheduling challenges. At the organizational level, three healthcare leaders were interviewed from five invited participants, while at the policy level, two policymakers/urban planners participated out of three invited. No refusals were related to the study topic. Sample size was guided by the principle of data saturation, defined as the point at which no new themes or substantive insights emerged from successive interviews or discussions. Saturation for FGDs was observed by the fifth and sixth group discussions, as recurring patterns regarding time constraints, workplace culture, and environmental barriers consistently emerged. For IDIs and KIIs, saturation was achieved when interviews yielded confirmatory rather than novel perspectives across ecological levels. Recruitment ceased once thematic saturation was reached and sufficient depth across stakeholder groups had been achieved.

## Data collection

Data collection was conducted using a qualitative approach comprising focus group discussions (FGDs), in-depth interviews (IDIs), and key informant interviews (KIIs) to explore multilevel determinants of physical activity, sedentariness, and exercise adoption. A hierarchical coding tree and semi-structured interview guides were developed based on the Social Ecological Model to systematically explore individual, interpersonal, community, organisational, and policy-level influences, with the coding framework refined inductively during analysis. FGDs were conducted to elicit shared experiences, social norms, and collective perceptions regarding physical activity and sedentary behaviour. Each FGD consisted of 6–9 participants with similar professional backgrounds to promote homogeneity and facilitate open discussion. Sessions were held in quiet, neutral venues that ensured privacy and minimal interruptions. At the beginning of each FGD, participants were introduced to the study objectives, ground rules were established (including confidentiality, respect for differing views, and voluntary participation), and rapport was built before discussions commenced. The moderator guided discussions using open-ended questions and follow-up probes, encouraged participation from all group members, and managed group dynamics to minimise dominance by individual participants.

IDIs were conducted with selected participants' family members to gain deeper insight into experiences, motivations, beliefs, and contextual factors influencing physical activity and exercise behaviours of HCPs. Interviews were conducted at locations convenient to participants or via secure virtual platforms where necessary. KIIs were conducted with purposively selected individuals based on their professional roles and policy-related decision-making. These interviews focused on organisational practices, institutional barriers, and policy-level influences affecting physical activity and exercise adoption. All FGDs, IDIs, and KIIs were conducted by the lead researcher, a public health physician trained in qualitative research methods. Data collection was conducted in English only and no interpreters were used. All sessions were audio-recorded with participants' verbal consent and lasted approximately 45–90 minutes. Participants were informed of their right to decline answering any question or withdraw from the study at any time without consequences. Field notes were taken during and immediately after each session to document contextual information, non-verbal cues, and reflexive observations. Data collection and analysis were conducted concurrently. Thematic saturation was considered achieved when successive FGDs, IDIs, and KIIs yielded no new codes, themes, or substantive insights across the levels of the Social Ecological Model. At this point, further data collection was deemed unlikely to add analytical value and recruitment was discontinued.

## Data analysis

All focus group discussions (FGDs), in-depth interviews (IDIs), and key informant interviews (KIIs) were conducted by the lead researcher, a public health physician with formal training in qualitative research methods and thematic analysis. The researcher had no prior personal or professional relationships with study participants, and no authority or supervisory roles existed between the researcher and participants at the time of data collection. This positionality was explicitly acknowledged throughout the research process to minimise potential power imbalances and interviewer influence. All data were anonymized by removing identifying information. Data analysis followed a thematic analysis approach, guided by the Social Ecological Model and informed by both deductive and inductive coding. An initial coding framework was developed deductively based on the study objectives and the SEM. Codes were grouped into themes and subthemes reflecting contributors, barriers, and facilitators of physical activity, sedentariness and exercise adoption. Representative participant quotes were used to illustrate findings. Coding was primarily conducted by the lead researcher, who has formal training in qualitative research methods and thematic analysis, as well as professional certification in group fitness instruction and licensure as a female Zumba instructor, providing contextual familiarity with physical activity behaviors and exercise environments.

To enhance analytic rigor and credibility, supervisors experienced in qualitative research provided expert review of the analytic process. This included examination of the coding structure, thematic development, and interpretive coherence, ensuring that findings were grounded in the data and appropriately contextualized. This review informed refinement of themes and strengthened the overall analytic quality of the research output. Dovetail qualitative analysis software was used to support transcription, data organization, code management, and systematic retrieval and comparison of data segments. In parallel, manual analytic techniques, including memo writing, code mapping, and theme clustering, were employed to facilitate reflexive engagement with the data and iterative refinement of themes. Data collection was conducted solely in English and no interpreters were required. Formal member checking with participants was not conducted due to logistical constraints. However, trustworthiness of the findings was enhanced through iterative reflexive analysis, peer debriefing within the research team, use of verbatim quotations to support emergent themes, and maintenance of a detailed audit trail documenting analytic decisions.

## Ethical approval

Ethical approval for this study was obtained from the Health Research and Ethics Committee (HREC) of Lagos University Teaching Hospital (HREC No: ADM/DSCST/HREC/APP/4115). Written informed consent was obtained from all

participants before participation. All procedures were conducted in accordance with the ethical standards of the institutional research committee and the 1964 Helsinki Declaration and its later amendments.

## Results

Table 1 summarizes the characteristics of the focus group participants. Mean age of participants was 38.7 (standard deviation 9.2) years and respondents were 87.2% female. Out of a total of 39 FGD participants, nurses made up 30.1%, consultant physicians made up 23.8%, and resident doctors made up 15.4% of the total. Community Health Officers (CHOs) and Community Healthy Extension Workers (CHEWs) made up 18.0% and 12.8% respectively. Table 2 presents the characteristics of respondents who participated in the in-depth and key informant interviews. Among the in-depth interview participants, all were male, ranging in age from 28 to 56 years and had known their HCP family member for between 10 and 28 years. Their relationship to healthcare professionals (HCPs) was primarily spousal, except for one respondent who was a sibling. For the organizational-level key informants, three senior managers were interviewed: two medical directors of general hospitals (one male and one female, aged 45 and 59 years, with 5 and 12 years in their roles, respectively) and one male Chairman of the Medical Advisory Council (aged 51 years, 5 years in position) from a teaching hospital. At the policy-making level, two female respondents were interviewed: a 45-year-old Chief Town Planning Officer with 4 years of experience in her role and a 46-year-old District Director of Medical Services and Disease Control with 2 years in position. Together, these respondents provided perspectives across interpersonal, organizational, and policy-making levels.

### Focus group discussions (Individual level)

**Theme 1: Personal experiences with physical activity, sedentariness, and exercise adoption.** Across focus groups, healthcare professionals (HCPs) described fluctuating, often declining engagement in structured exercise as careers and family responsibilities intensified. Several participants traced a trajectory from youthful activity to mid-career lapses: *"As the years go by, the level of exercising has reduced… Maybe there's a need for more intentionality in ensuring that exercise goes on"* (Consultant, M, 47). Competing priorities, fatigue, and limited supportive environments were recurrent. One Senior Registrar summarized the shift succinctly: *"As I grew older, life grew more complex. One thing or the other just gets in the way and so I just leave exercise to stretches"* (F, 41). Others reported repeated attempts without sustained adherence: *"Exercise is a no-no for me… I tried several times, then stopped"* (Consultant, F, 45).

At the same time, many participants reported intentional micro-behaviours to counter long sitting, such as scheduled stand breaks during clinics and purposeful movement between departments: *"I consciously try to not sit down for prolonged periods without standing up for a five-minute break… I try to encourage others to do the same thing"* (Consultant, F, 52). Nurses frequently described high incidental activity embedded in their roles—walking, lifting, and multitasking—but difficulty carving out time for structured exercise: *"Not during work time, but during leave… I do only what I can"* (Nurse, F, 55). Others leveraged home-based strategies (videos, stair-climbing, weekend jogging) and active commuting: *"I decided to park [the car] and take a walk from my house to the bus stop… because I was actually adding weight"* (Nurse, F, 42). Several reported tangible benefits, including weight loss following step-count goals: *"Then at home, I do weekend jogging… I started with 1,500 steps… I lost 14 kg"* (Nurse, F, 35).

**Theme 2: Contributors to physical inactivity, sedentariness, and exercise non-adoption.** *Subtheme 2.1: Everyday drivers of sedentariness*

Participants cited cumulative fatigue after long shifts, screen time, and prolonged commuting as primary drivers of sedentary time. Social media and passive entertainment filled scarce downtime: *"You can sit down and be on Instagram scrolling through for 2 hours… you're being sedentary"* (CHO, F, 53). Lagos traffic was described as "enforced sitting," particularly for those reliant on public transport. Economic stressors (poverty, unemployment) and psychosocial strain were also cited: *"Even an unhappy marriage can also cause it"* (CHEW, F, 30).
*Subtheme 2.2: Built environment and safety*

**Table 1. Characteristics of Focus Group Discussion (FGD) participants.**

| FGD Nos. | Participants | Sex | Age | Exercise adoption | Designation |
|---|---|---|---|---|---|
| FGD 1 | Speaker A | Female | 41 | Adopter | Senior Registrar |
| | Speaker B | Female | 35 | Adopter | Senior Registrar |
| | Speaker C | Male | 31 | Adopter | Junior Registrar |
| | Speaker D | Female | 39 | Adopter | Senior Registrar |
| | Speaker E | Male | 40 | Adopter | Senior Registrar |
| | Speaker F | Female | 38 | Adopter | Junior Registrar |
| FGD 2 | Speaker A | Female | 49 | Non-Adopter | Consultant |
| | Speaker B | Female | 45 | Non-Adopter | Consultant |
| | Speaker C | Female | 52 | Non-Adopter | Consultant |
| | Speaker D | Male | 47 | Non-Adopter | Consultant |
| | Speaker E | Female | 50 | Non-Adopter | Consultant |
| | Speaker F | Male | 42 | Non-Adopter | Consultant |
| | Speaker G | Female | 46 | Non-Adopter | Consultant |
| | Speaker H | Female | 45 | Non-Adopter | Consultant |
| | Speaker I | Female | 42 | Non-Adopter | Consultant |
| FGD 3 | Speaker A | Female | 53 | Non-Adopter | Nurse |
| | Speaker B | Female | 35 | Non-Adopter | Nurse |
| | Speaker C | Female | 25 | Non-Adopter | Nurse |
| | Speaker D | Female | 32 | Non-Adopter | Nurse |
| | Speaker E | Female | 38 | Non-Adopter | Nurse |
| | Speaker F | Female | 55 | Non-Adopter | Nurse |
| FGD 4 | Speaker A | Female | 40 | Adopter | Nurse |
| | Speaker B | Female | 38 | Adopter | Nurse |
| | Speaker C | Female | 31 | Adopter | Nurse |
| | Speaker D | Female | 35 | Adopter | Nurse |
| | Speaker E | Female | 39 | Adopter | Nurse |
| | Speaker F | Female | 42 | Adopter | Nurse |
| FGD 5 | Speaker A | Female | 53 | Adopter | CHO* |
| | Speaker B | Female | 29 | Adopter | CHO |
| | Speaker C | Female | 22 | Adopter | CHEW** |
| | Speaker D | Female | 30 | Adopter | CHO |
| | Speaker E | Female | 48 | Adopter | CHEW |
| | Speaker F | Male | 43 | Adopter | CHO |
| FGD 6 | Speaker A | Female | 26 | Non-adopter | CHO |
| | Speaker B | Female | 29 | Non-adopter | CHEW |
| | Speaker C | Female | 27 | Non-adopter | CHEW |
| | Speaker D | Female | 35 | Non-adopter | CHEW |
| | Speaker E | Female | 32 | Non-adopter | CHO |
| | Speaker F | Female | 30 | Non-adopter | CHO |

*Community Health Officer **Community Health Extension Worker; Mean age 38.7(sd9.2); Females 82.7%, Males 17.3%.

The urban context emerged as a dominant constraint. Participants described narrow or broken sidewalks, vehicles encroaching on walkways, encumbered public spaces, and scarce, unsafe parks: *"Even if you have sidewalks, cars are passing on the sidewalks… you can easily be knocked down"* (Consultant, M, 31). Safety concerns—poor lighting,

**Table 2. Characteristics of in-depth and key informant interview respondents.**

| Interviews | Respondents | Age | Sex | Years in position | Relationship to HCPs/managerial position |
|---|---|---|---|---|---|
| In-Depth Interviews Interpersonal | | | | | |
| | Respondent 1 | 56 | Male | 10* | Spouse |
| | Respondent 2 | 45 | Male | 12* | Spouse |
| | Respondent 3 | 47 | Male | 14* | Spouse |
| | Respondent 4 | 28 | Male | 28* | Sibling |
| Key Informant Interviews Organizations | | | | | |
| | Manager 1 | 45 | Male | 5 | Medical Director – General Hospital |
| | Manager 2 | 59 | Female | 12 | Medical Director – General Hospital |
| | Manager 3 | 51 | Male | 5 | Chairman Medical Advisory Council (CMAC) – Teaching Hospital |
| Key Informant Interviews Policy Makers | | | | | |
| | Policymaker 1 | 45 | Female | 4 | Chief Town Planning Officer |
| | Policymaker 2 | 46 | Female | 2 | District Director, Medical Services and Disease Control |

*Years of relationship.

early-morning and evening insecurity—discouraged outdoor exercise: *"Security is a major issue in my area"* (Consultant, F, 52); *"Going out to jog… it's not safe for me in my environment"* (Senior Registrar, F, 42). Air and noise pollution, flooding, and harsh sunlight were additional deterrents. One nurse added: *"Even the few parks we have are occupied by 'agberos'… The little walkways are covered by sellers of goods"* (F, 35).

*Subtheme 2.3: Culture, social norms, and religion*

Cultural norms shaped attitudes toward movement and body size. Active transport (trekking) was viewed by some as a marker of lower status: *"The mentality is that poor people walk, the rich drive"* (Senior Registrar, M, 40). Several participants noted that being "full" or overweight is sometimes praised—*"We appreciate, especially for women, when they are looking 'full'"* (CHEW, F, 27)—and that weight loss can attract negative speculation (e.g., marital problems or illness). Post-partum practices emphasizing caloric loading and rest, and traditions such as "fattening rooms," were seen as reinforcing sedentariness: *"Some cultures have fattening rooms. It's part of our culture"* (Nurse, F, 53). Modesty norms, particularly for women who wear hijab, constrained clothing options and comfort for outdoor exercise: *"…don't expect them to exercise, except if it's indoor"* (Nurse, F, 31). Despite these barriers, participants observed a gradual shift toward acceptance of exercise (e.g., church aerobics), especially among younger people: *"People are beginning to imbibe exercise as a way of life"* (Consultant, F, 45).

*Subtheme 2.4: Healthcare organizational influences*

Workplace environments varied widely. Some organizations adopted nudges (e.g., stair prompts, disabling lifts temporarily) and permitted exercise breaks; others offered no structured supports. Staff shortages intensified workload and limited flexibility for breaks: *"Everybody is on his or her toes… You must get up to do one thing or the other for the work to be done"* (Nurse, F, 42). Participants emphasized that institutional supports matter but cannot substitute for personal agency: *"The work environment has a role to play, [but] individual decision is still a big factor"* (Senior Registrar, M, 40).

**Theme 3: Facilitators of physical activity and exercise adoption.** Facilitators spanned intrinsic motivation, health symptoms, and social/organizational supports. Many were motivated by weight management, longevity, and symptom relief: *"I started having cramps on my knees… it was health-related"* (Nurse, F, 38). Visible role models and peer accountability helped sustain routines: *"When you look at someone that exercises, you have this feeling this person is*

living right" (CHEW, F, 32); *"My success story is when I teamed up with my sister and hired a gym instructor. Even when he stopped, we were able to continue"* (Senior Registrar, F, 38). Participants repeatedly called for safe environments and institutional facilitation—onsite gyms, equipment access, structured classes, and designated time: *"If I live in a safe environment, I will take regular walks"* (Senior Registrar, F, 41).

**Theme 4: Barriers across specialties and cadres.** Participants described specialty-specific sedentariness (e.g., prolonged seated consultations; laboratory review) and cadre-related differences, with senior staff reporting more sitting and junior cadres more incidental movement but less time for structured exercise. Time scarcity, emotional stress, and family responsibilities (childcare, cooking) constrained adherence: *"You get home after much work; you still go and cook… And at home again, the same stress"* (Nurse, F, 32). Environmental barriers (distance/cost of gyms) and individual factors (procrastination, lack of a workout partner) compounded these constraints: *"It helps when you have people that can encourage you"* (Nurse, F, 40).

**Theme 5: Adaptation strategies to overcome barriers.** HCPs described pragmatic adaptations: short indoor routines, video-guided sessions, stair-climbing, and evening walks/jogs—sometimes with family for accountability. Intentional goal-setting (e.g., BP monitoring), reframing constraints, and building habits were emphasized: *"The environment may not be so conducive… but I shouldn't be using that as an excuse… I just need more motivation"* (FGD2, F, 45). Participants highlighted that small, consistent actions were more sustainable than ambitious but sporadic workouts.

**Theme 6: Participant-generated recommendations for organizations and systems.** Participants proposed comprehensive, multi-level strategies. They emphasized infrastructure and time: *"They can create a gym in our healthcare facilities if they want us to exercise so that we can go there when we are not busy"* (Nurse, F, 25). Incentives were also recommended: *"Incentives always work… Something as small as bread can motivate people"* (Senior Registrar, F, 41). Leadership and role-modelling were viewed as crucial: *"My boss, she's very active, very agile. Seeing her doing many things… encourages us"* (CHEW, F, 35). Participants also stressed staffing adjustments to allow exercise breaks: *"There have to be more health workers on the ground… You can cover up and then maybe when it's your turn, I can always cover up for you while you go"* (Nurse, F, 53). They called for widespread awareness and education: *"Everybody should have the exercise mentality… From the community level to the states to the top"* (Nurse, F, 35).

### Synthesis

In summary, participants demonstrated high occupational movement but irregular structured exercise, constrained by time, safety, infrastructure deficits, cultural norms, modesty considerations, and variable organizational supports. Facilitators clustered around personal health goals, peer and leadership modeling, safe spaces, and organizational provision of time and resources. Adaptation strategies were practical and home- or facility-based, relying on accountability and routine. Recommendations consistently endorsed multi-level, context-specific interventions aligned with the Social Ecological Model, emphasizing the need to (1) improve environmental safety and access, (2) embed exercise into organizational culture and scheduling, (3) leverage leadership and peer influence, and (4) address cultural and gender-specific barriers through sensitive programming and communication.

### In-Depth interview report (Interpersonal level)

**Theme 1: Perception of physical activity, sedentariness, and exercise adoption of HCPs.** The prevalence of physical inactivity among healthcare professionals was highlighted across all interviews. While the professionals understand the benefits of regular exercise, most do not engage consistently. Respondent 2, for instance, described their healthcare professional spouse as having a sedentary lifestyle despite a supportive environment and occasional encouragement to exercise. Respondent 4 shared that their healthcare professional relative was motivated to begin exercising after purchasing a treadmill, yet they have not sustained regular activity levels. In many cases, physical inactivity appeared to be linked to work-related fatigue, as noted by one respondent: *"She knows the benefits of exercise,*

*but after a long day at work, she feels too drained to do anything."* (Respondent 1, Male, 56 years) Sedentariness is a significant concern, with participants reporting long periods of sedentariness during both work and leisure. Healthcare professionals often spend their free time engaged in passive activities such as phone usage, watching television, or resting after shifts. Respondent 3 observed that their healthcare professional relative has a sedentary lifestyle at home, stating: *"After work, she prefers to rest or use her phone instead of exercising."* (Respondent 3, Male, 47 years) Respondent 2 noted that even at work, healthcare professionals may experience prolonged periods of standing or sitting, further contributing to their overall sedentary behaviour. Respondents believed that the physical demands of healthcare work, coupled with limited energy during downtime, exacerbate sedentariness.

**Theme 2: Contributors to physical inactivity, sedentariness and exercise non-adoption.** Work schedules and family responsibilities were identified as key deterrents to physical activity. They bemoaned the fact that their healthcare professional relatives often work long hours under stressful conditions, leaving little time or energy for exercise. Respondent 3 explained that balancing family duties with work responsibilities creates significant challenges, stating: *"She barely has time for herself, let alone physical activity."* (Respondent 3, Male, 47 years) Fatigue and low energy levels were recurring themes across interviews, with Respondent 2 noting that while their spouse has time for exercise, *"Mental exhaustion from work makes it hard for her to feel motivated to exercise."* (Respondent 2, Male, 45 years) Lack of motivation among their HCP family members was another issue frequently highlighted by respondents. Respondent 4 mentioned that their relative, despite wanting to be more active, often struggles with initiating or maintaining regular exercise routines. The influence of cultural and social on physical activity behaviours was also alluded to. While none of the respondents identified cultural barriers to exercise, Respondent 2 remarked that physical activity is not traditionally emphasised in their community. However, Respondent 4 pointed out that their family's cultural practices during festivals, which involve active participation in events, positively influence physical activity habits and promote active lifestyles. Social factors also play a role. The absence of peers or colleagues who prioritise physical activity was noted as a potential deterrent. One respondent observed:*"The healthcare professional does not have friends who regularly exercise, which might contribute to her physical inactivity."* (Respondent 2, Male, 45 years) The consensus was that the lack of socially active peer groups may limit opportunities for healthcare professionals to engage in group fitness activities or receive encouragement from their immediate social network.

**Theme 3: Facilitators of physical activity, non-sedentariness and exercise adoption.** Despite the challenges, several facilitators of exercise adoption were identified. Family support emerged as a critical enabler. Respondent 4 explained that their family regularly invites the healthcare professional to participate in activities such as jogging, gym sessions, and walking, which have contributed to improving her physical strength. Spousal support was particularly effective, with Respondent 1 noting that shared activities, such as gym visits or evening walks, have helped their spouse remain somewhat active. Cultural norms, while not universally influential, were also noted to serve as facilitators. Respondent 4 shared that their family's traditions of incorporating physical activities into festivals encourage a mindset where exercise is seen as enjoyable and meaningful. Flexible work hours, where available, were another facilitator mentioned by respondent. Respondent 2 suggested that adjusted schedules could create more time for physical activities, especially for professionals with demanding workloads. Community programmes designed to promote physical activity were identified as valuable resources. Respondent 1 proposed hospital-organised initiatives to educate healthcare professionals on the importance of fitness while also reducing workload pressures. They believed that these programmes, combined with family encouragement and social support, could provide a comprehensive framework for fostering exercise adoption.

**Theme 4: Barriers to physical activity, non-sedentariness and exercise adoption.** Barriers to exercise adoption were consistently highlighted across all interviews. Time constraints due to long work hours and family responsibilities were the most significant obstacles. Respondent 3 emphasised that healthcare professionals often prioritise work and caregiving roles over self-care, stating:*"It's not just about having time; they also need energy and the right mindset to*

*engage in physical activity.*" (Respondent 3, Male, 47 years) Fatigue and low energy levels further exacerbate this issue, with Respondent 2 describing how physical and emotional exhaustion often deter their spouse from exercising, even when time is available. Motivational challenges were also identified as impeding regular exercise. Respondent 4 noted: "*She wants to be more active but struggles with the discipline to stick to a routine.*" (Respondent 4, Male, 28 years) Limited access to gym facilities or suitable environments for outdoor exercise was occasionally mentioned as a barrier. Respondent 1 added that the availability of fitness centres could influence participation, particularly for those who require structured settings for exercise.

Theme 5: Recommendations for supporting exercise adoption and physical activity. The findings highlight the importance of addressing both structural and interpersonal factors to support healthcare professionals in adopting regular physical activity. Families can play a significant role by creating an environment conducive to exercise. Respondent 4 suggested organising group activities, setting shared fitness goals, and offering assistance with household responsibilities to free up time for physical activity. Hospitals and workplaces could also implement flexible schedules and wellness programmes that prioritise fitness. Respondent 1 proposed that hospitals organise initiatives to reduce workload pressures, stating: "*Such programmes would help healthcare professionals find time for physical fitness.*" (Respondent 1, Male, 56 years) Promoting peer influence and social engagement was another key recommendation. Respondent 2 highlighted the potential of reconnecting healthcare professionals with physically active colleagues or friends to foster a sense of camaraderie and motivation. Community-based fitness programmes tailored to healthcare professionals' schedules could further encourage participation. Finally, they proposed that cultural and social frameworks should be leveraged to normalise and encourage physical activity. Respondent 4's experience with culturally embedded physical activities illustrates the potential of integrating exercise into social events and traditions.

### Synthesis

The interviews revealed a complex interplay of factors influencing physical inactivity, sedentariness, and exercise adoption among healthcare professionals in Lagos State from the perspectives of their closest family members. While barriers such as demanding work schedules, fatigue, and motivational challenges persist, facilitators such as family support, spousal involvement, and cultural influences offer pathways to improvement. By addressing these barriers and leveraging existing facilitators, healthcare professionals can be better supported to adopt and maintain regular physical activity, contributing to improved physical and mental wellbeing.

### Key informant interview report – Organisational level

Theme 1: Perception of physical activity, sedentariness, and exercise adoption of HCPs. All participants emphasised the widespread prevalence of physical inactivity among healthcare professionals in Lagos State. The primary contributors were high workloads and time constraints. One manager stated: "*Our staff are so overburdened with their patient loads that exercise becomes an afterthought. In some clinics, doctors see over 30 patients daily. Where would they find the time for physical activity?*" Manager 1, Male, 45 years Another manager highlighted a similar trend: "*Since the facility's renovation increased our patient capacity, the workload has skyrocketed, leaving little time for staff to prioritise exercise.*" Manager 2, Female, 59 years Another manager pointed to the post-COVID-19 workload as a significant driver of inactivity: "*The pandemic created a situation where staff were stretched to their limits. Even now, we're still understaffed, which means fewer people are doing more work, and physical activity is one of the first things they abandon.*" Manager 3, Male, 51 years The same manager also pointed to the persistent "japa" syndrome as responsible for the dearth of healthcare professionals, compounded by low recruitment numbers due to inadequate human resource budgets. In all, responses indicated that while some healthcare professionals might be aware of the benefits of exercise, the high demands of their jobs significantly limit their ability to engage in physical activity. Sedentariness was

particularly noted among non-surgical and administrative healthcare professionals. The long hours of sitting and lack of structured interventions to combat this behavior have exacerbated the problem according to the respondents. One manager observed: *"In outpatient and administrative roles, healthcare professionals spend most of their shifts seated. This sedentary lifestyle is as much of a problem as the lack of exercise itself." Manager 3, Male, 51 years* One manager added:*"Even roles that involve some movement are not active enough to combat the negative effects of sedentariness. Without deliberate interventions, these habits persist." Manager 2, Female, 59 years*

**Theme 2: Contributors to physical inactivity, sedentariness and exercise non-adoption.** The most significant determinant of inactivity across all interviews was the demanding workload and inadequate staffing levels. One manager noted: *"The sheer number of patients our staff are expected to handle daily leaves them drained. It's not just about time—it's the exhaustion they feel after their shifts." Manager 1, Male, 45 years "We're struggling with understaffing, and this means healthcare professionals are multitasking and working longer hours. It's hard to think about exercise when you barely have time for a break." Manager 2, Female, 59 years* The lack of designated spaces for physical activity was another factor identified. *"The renovations improved patient care spaces but didn't consider staff wellness. We don't have a dedicated gym or even a proper area where staff can unwind or exercise." Manager 2, Female, 59 years* Another manager echoed this sentiment: *"While we've set aside some space for activities, it's not equipped to meet the needs of a diverse group of professionals." Manager 1, Male, 45 years* The managers also noted that physical activity was not deeply embedded in the organisational culture of their organisations.*"There's no policy or systemic encouragement for exercise. We have to change the mindset and create a culture where physical activity is seen as essential, not optional." Manager 3, Male, 51 years*

**Theme 3: Facilitators of physical activity, non-sedentariness and exercise adoption.** The participants proposed several strategies to encourage exercise adoption among healthcare professionals. Increasing awareness of the benefits of exercise and the risks of inactivity was a recurring theme.*"A lot of healthcare workers know exercise is good for them, but consistent reminders through wellness talks or campaigns can help keep it on their radar." Manager 3, Male, 51 years* Appointing leaders within departments to promote exercise as exercise champions was seen as a likely facilitator.*"If every department had a champion for physical activity, it would create a ripple effect. People follow what they see their leaders doing." Manager 1, Male, 45 years* Using fitness trackers and mobile apps to encourage physical activity and track progress was suggested. *"These tools can make exercise engaging. Imagine friendly competitions among departments based on step counts - this can motivate even the busiest professionals." Manager 3, Male, 51 years* Collaboration with external organisations was another idea.*"Registering staff at clubs for activities like swimming or tennis, with discounts for group registrations, could make exercise more accessible." Manager 2, Female, 59 years*

**Theme 4: Barriers to physical activity, non-sedentariness and exercise adoption.** The most frequently mentioned barrier was the lack of time and energy due to demanding work schedules.*"The long shifts and high patient load leave staff physically and mentally drained. Even when they have time, they don't have the energy to exercise." Manager 1, Male, 41 years "It's not that they don't want to exercise; it's that they are completely exhausted by the time their shifts end." Manager 2, Female, 59 years* The absence of policies and structured programs to promote physical activity was another major barrier.*"Without official backing, these initiatives feel like afterthoughts. We need clear policies to prioritise staff wellness." Manager 3, Male, 51 years* Another manager added: *"Even if we want to promote exercise, the lack of resources—like a proper gym—limits what we can achieve." Manager 2, Female, 59 years*

**Theme 5: Recommendations for supporting exercise adoption and physical activity.** Incorporating brief physical activity breaks during work shifts was a recommended solution to encourage healthcare professionals to engage in exercise without disrupting their demanding schedules. These short, structured activities could provide a practical solution to combat sedentariness and improve overall well-being in a high-pressure work environment. One manager highlighted this strategy, stating: *"Even a five-minute stretch or walk during breaks can make a difference." Manager 2, Female. 59 years* Leveraging technology and gamification was another recommendation. By introducing fitness apps and trackers,

healthcare facilities can foster a culture of healthy competition and engagement among staff. One manager emphasised the potential of this approach, noting: *"Staff are more likely to engage when there's an element of fun or competition."* *Manager 3, Male, 51 years* Such tools can track activity levels, encourage goal setting, and create camaraderie among colleagues through friendly competitions. Recognition and incentive programs were also suggested as tools that can play a crucial role in motivating staff to adopt an active lifestyle. One manager suggested rewarding participants with tokens of appreciation, saying: *"Incentives like recognition plaques or even small gifts can motivate people to be more active."* *Manager 1, Male, 45 years* These programs could reinforce positive behaviour and help integrate physical activity into the organisational culture. In addition, optimising existing resources was seen as a practical approach to overcoming space and equipment constraints. One manager pointed out that even with limited facilities, creativity can drive change: *"We don't need fancy gyms; we just need to creatively use what we have."* *Manager 2, Female, 59 years* They suggested that facilities like physiotherapy departments and underutilised spaces can be repurposed for structured exercise programs. Finally, advocating for policy support was identified as essential to ensuring sustainability and institutionalisation of wellness programs. One manager highlighted the importance of this step, stating: *"Policy support is essential to making these changes sustainable."* *Manager 3, Male, 51 years* The respondents agreed that healthcare facilities can establish formal wellness initiatives, creating environments that prioritise the health and well-being of their staff.

### Synthesis

This report highlights the complex interplay of workload, organisational culture, and resource limitations in shaping physical inactivity and sedentariness among healthcare professionals in Lagos State. While significant barriers exist, there are also practical facilitators, such as awareness campaigns, leadership champions, and technological tools, that can drive change. By addressing these issues through targeted interventions and robust policy support, healthcare facilities can foster a culture of physical activity, improving both the health and job satisfaction of their staff.

### Key informant interview report – Policy level

This report synthesizes insights from a key informant interview with two respondents: the Chief Town Planning Officer of Lagos State and the Director of Medical Services for District 6 of the Lagos State Ministry of Health. The report is structured according to the study's research objectives to explore the determinants of physical inactivity, sedentariness, and exercise adoption among healthcare professionals.

**Theme 1: Perception of physical activity, sedentariness, and exercise adoption of HCPs.** Physical inactivity among healthcare professionals in Lagos State is prevalent, as indicated by both respondents. Policymaker 1 noted that while some healthcare professionals are physically active, many struggle to prioritise exercise due to demanding work schedules and the absence of recreational facilities within healthcare establishments. Policymaker 2 corroborated this, highlighting that existing policies mandate monthly physical activity for civil servants, including healthcare professionals. However, gaps in implementation and enforcement lead to inconsistent participation. *"Although healthcare professionals understand the importance of physical activity, they often neglect it due to the pressures of their work and limited opportunities to exercise regularly."* *(Policymaker 2, Female, 46 years)* This pattern highlights a disconnect between awareness and practice, emphasising the need for practical interventions. Sedentariness was another significant concern, driven by both professional and urban constraints. Policymaker 1 explained that healthcare professionals in Lagos are often subjected to sedentary behaviours due to long work hours and the design of their workspaces, which do not encourage movement. This is exacerbated by the physical environment in Lagos, where traffic congestion and inadequate public transport systems force healthcare workers into prolonged periods of inactivity during commutes. Policymaker 2 noted that healthcare professionals also face a lack of designated spaces for recreational activities in their workplaces. *"Even when professionals are willing to be active, the environment does not support it. Facilities are either absent or inaccessible."* *(Policymaker 2, Female, 46 years)* They agreed that a combination of structural and professional barriers perpetuates sedentary lifestyles among healthcare professionals.

**Theme 2: Contributors to physical inactivity, sedentariness and exercise non-adoption.** Several determinants of physical inactivity and sedentariness emerged during the interviews. Workload emerged as the most significant factor, with both respondents identifying overwork and understaffing as key issues. Policymaker 1 remarked: *"Healthcare professionals are so consumed by their duties that physical activity often becomes a secondary priority." Policymaker 1, Female, 45 years* Urban planning issues were also identified as determinants. Policymaker 1 highlighted that older parts of Lagos lack sufficient pedestrian-friendly areas and recreational spaces, which discourages active lifestyles. Although newer developments incorporate public spaces and parks, their accessibility remains limited, particularly for those working in high-density areas. Safety concerns, including poor lighting and encroachment on walkways, further deter outdoor exercise. Policymaker 2 added that institutional policies within healthcare facilities rarely prioritise physical activity. *"While there are general mandates for civil servants, specific efforts targeting healthcare professionals remain lacking. Without tailored policies, it is difficult to address their unique challenges." (Policymaker 2, Female, 46 years)*

**Theme 3: Facilitators of physical activity, non-sedentariness and exercise adoption.** Despite the barriers, several facilitators of exercise adoption were discussed. Both respondents agreed that a supportive work environment and accessible recreational infrastructure could significantly enhance physical activity levels among healthcare professionals. Policymaker 1 pointed to ongoing efforts to integrate fitness-friendly infrastructure into Lagos's urban planning. *"Our master plan includes provisions for one stadium per local government area. Although implementation is slow, these spaces could become valuable resources for healthcare professionals." (Policymaker 2, Female, 46 years)* Policymaker 1 emphasised the importance of workplace wellness programmes. She suggested incorporating structured physical activities into healthcare institutions' schedules and leveraging national health awareness days to promote exercise. *"Healthcare facilities should be champions of active lifestyles. Organising regular fitness events can encourage participation and create a culture of wellness." (Policymaker 1, Female, 45 years)* Both respondents also recognised the potential of community-based initiatives, such as exercise groups and public health campaigns, to facilitate exercise adoption. These programmes could address broader structural barriers while fostering a sense of collective motivation among healthcare professionals.

**Theme 4: Barriers to physical activity, non-sedentariness and exercise adoption.** Numerous barriers to exercise adoption were identified, spanning professional, structural, and policy-related domains. The most significant professional barrier is workload. *"Healthcare professionals face immense demands, leaving them physically and mentally drained by the end of the day. Exercise often becomes an afterthought." (Policymaker 2, Female, 46 years)* Urban barriers, including traffic congestion and poor infrastructure, were also highlighted. Policymaker 1 observed: *"Even if healthcare professionals are motivated to exercise, navigating Lagos's traffic and finding safe spaces to do so can be overwhelming." (Policymaker 1, Female, 45 years)* The lack of pedestrian walkways, insufficient lighting, and air pollution were additional challenges cited. Policy implementation issues further compound these barriers. Both respondents agreed that bureaucratic delays, funding constraints, and competing priorities hinder the development and enforcement of physical activity policies. *"Leadership must take a proactive approach. Without sustained advocacy, these initiatives will remain on paper." (Policymaker 2, Female, 46 years)*

**Theme 5: Recommendations for supporting exercise adoption and physical activity.** Both respondents provided actionable recommendations to address physical inactivity among healthcare professionals. Policymaker 1 emphasised the need to prioritise fitness-friendly urban infrastructure, suggesting that policies should allocate resources for parks, fitness centres, and pedestrian-friendly spaces. She added that early budget proposals could help address funding constraints and ensure timely implementation. Policymakers 2 recommended that healthcare institutions adopt written workplace policies to combat sedentary behaviours. She suggested integrating physical activity into daily routines through initiatives like fitness breaks or after-work exercise classes. She also proposed tax incentives for private fitness centres to increase accessibility for healthcare professionals. Both respondents agreed on the importance of collaboration. *"Urban*

*planners, healthcare leaders, and public health officials must work together. Only through multisectoral partnerships can we create an environment that supports active lifestyles." (Policymaker 1, Female, 45 years)*

### Synthesis

This key informant interview report highlights the multifaceted nature of physical inactivity and sedentariness among healthcare professionals in Lagos State. While workload, urban constraints, and policy gaps remain significant challenges, there is considerable potential for improvement through targeted interventions. By enhancing urban infrastructure, implementing workplace wellness programmes, and fostering collaborative efforts, Lagos State can create an enabling environment for healthcare professionals to adopt and sustain active lifestyles. The insights provided by the respondents highlight the importance of integrating policy changes with structural improvements to address physical inactivity comprehensively.

### Summary of in-depth interviews and key informant interviews

The in-depth interviews and key informant interviews with family members of healthcare professionals, healthcare facility leaders, and policymakers in Lagos State revealed a multifaceted and complex interplay of factors influencing physical inactivity, sedentariness, and exercise adoption. Workload demands, organisational culture, urban constraints, and policy gaps emerged as significant barriers to physical activity, compounded by fatigue and motivational challenges. Healthcare professionals face long hours, limited resources, and a lack of structured wellness programmes, which exacerbate sedentariness and limit opportunities for exercise. Despite these barriers, the interviews also highlighted several facilitators that offer pathways to improvement. Family support, spousal involvement, cultural influences, and departmental champions provide social and interpersonal frameworks to encourage physical activity. Within organisations, awareness campaigns, leadership champions, and technological tools such as fitness trackers were identified as effective mechanisms for promoting exercise. Policymakers and healthcare leaders emphasised the importance of improving urban infrastructure, enhancing resource availability, and integrating workplace wellness programmes to address these challenges comprehensively. Enhancing organisational support, fostering collaborative efforts, and creating an enabling environment can were all identified as tools to drive meaningful change.

### Discussion

This qualitative study explored multilevel facilitators and barriers to physical activity (PA) among healthcare professionals (HCPs) in Lagos, aligning with the Social Ecological Model and mirroring patterns seen globally. Individually, lack of time, heavy workloads, and low self-efficacy constrained exercise despite high awareness – findings consistent with international evidence that long shifts and fatigue erode HCPs' health behaviours and that self-efficacy strongly predicts PA uptake. [10,11] Interpersonally, peer support and visible role-modelling by colleagues catalysed exercise, echoing studies showing that active clinicians are more likely to counsel – and be believed by – patients. [12,13] Organizationally, onsite facilities, protected activity breaks, and supportive policies enabled PA, whereas rigid schedules and staffing shortages impeded it – paralleling workplace-health-promotion reviews demonstrating that environmental and policy supports improve employee PA. [13,14] Environmental constraints – unsafe streets, poor lighting, encroached sidewalks, scarce parks, traffic, heat, and pollution – closely match evidence that built environments and perceived safety are decisive for routine PA. [15,16] Nigerian and broader African studies similarly link poor walkability and security concerns to lower activity. [6,17] Cultural norms that valorise larger body size – especially for women – and stigmatise active transport reproduced patterns reported in West and Southern Africa where body-image ideals and modesty norms can suppress women's PA. [18,19] In addition, participants' calls for safer public spaces, active-travel infrastructure, and mass-media campaigns align with the WHO "Health in All Policies" and Global Action Plan on Physical Activity, which emphasise cross-sector urban design and systems approaches. [20]

At the organisational and policy levels, the key informant interviews reinforced how excessive workloads, staff shortages, and weak institutional support structures perpetuate physical inactivity and sedentariness among health-care professionals in Lagos State. Managers' recognition of organisational culture as a determinant of exercise adoption echoes studies showing that leadership endorsement and visible managerial participation are critical for normalising physical activity within workplaces. [21,22] At the policy level, the lack of sustained implementation and enforcement of existing physical activity mandates mirrors findings from other African urban contexts, where policy fragmentation and competing health priorities undermine population-level exercise promotion. [6,23] Nonetheless, the facilitators identified – including departmental "exercise champions," wellness initiatives, and the integration of physical activity into staff schedules – demonstrate feasible strategies for change. Similar multi-component interven-tions in Ghana and Tanzania, combining leadership support, environmental restructuring, and flexible scheduling, have yielded measurable improvements in staff activity levels. [24] The insights from policymakers in this study also emphasise the importance of urban planning and intersectoral collaboration, reinforcing WHO's call for whole-of-government approaches to physical activity promotion through transport, urban design, and occupational health policies. [25] (Collectively, these findings highlight that addressing physical inactivity among healthcare professionals requires not only individual motivation and interpersonal support but also systemic institutional reforms and strong political commitment to embed physical activity into organisational and public-sector frameworks. Multi-level strat-egies that pair individual behaviour-change supports with organizational reforms and city-level infrastructure – and that are culturally sensitive and gender-responsive – are therefore most likely to shift norms and sustainably increase PA among HCPs and, by extension, the communities they serve. Our data reinforce that HCP physical inactivity undermines both staff health and the credibility and frequency of PA counselling.

## Study limitations

This study has several limitations that should be acknowledged. First, as a qualitative inquiry, findings are not statis-tically generalizable beyond the study population. Second, responses relied on self-report, which may be subject to recall and social desirability bias, particularly given cultural sensitivities surrounding physical activity and body image. Third, focus group dynamics may have influenced the openness of participants, with some voices dominating discus-sions and others possibly withholding perspectives. Fourth, the study was conducted within public healthcare facilities in Lagos State, and the experiences of healthcare professionals in private settings may differ. Finally, the study did not include longitudinal follow-up, limiting insights into how barriers and facilitators evolve over time. Despite these limitations, the study provides rich, contextual understanding of the multifaceted influences on physical activity among healthcare professionals.

## Recommendations

Findings from this study highlight the need for multi-level interventions. As indicated by FGD participants, at the organiza-tional level, healthcare facilities should adopt policies that promote physical activity by providing onsite gyms, incorporat-ing scheduled exercise breaks, and fostering workplace wellness cultures. Adequate staffing levels are critical to reduce workload pressures and enable time for physical activity. At the community level, investment in safe recreational spaces, pedestrian infrastructure, and improved security would create enabling environments for exercise. Policymakers should prioritize health in urban planning, develop supportive workplace wellness policies, and launch sustained public health campaigns to promote active lifestyles. Particular attention is needed to address the unique barriers faced by female healthcare professionals, such as cultural expectations and family responsibilities, through inclusive and flexible work arrangements. For policymakers in Lagos State, a deliberate focus on embedding physical activity promotion into health workforce wellness programs, protecting and expanding safe public exercise spaces through collaboration with urban planning and security agencies, and incorporating physical activity messaging into state-level campaigns aligned with

non-communicable disease prevention strategies will be crucial. Strengthening healthcare professionals as role models through leadership engagement, peer-led initiatives, and visible advocacy will further reinforce positive behaviors and amplify the public health impact of physical activity promotion.

## Conclusions

This qualitative study revealed that healthcare professionals in Lagos State face significant barriers to physical activity, including time constraints, heavy workloads, inadequate infrastructure, cultural norms, and safety concerns. Nevertheless, facilitators such as peer support, leadership role modeling, organizational policies, and personal motivation provide important opportunities for intervention. Addressing these barriers through comprehensive, multi-level strategies is critical, not only for improving the health of healthcare professionals themselves but also for enhancing their effectiveness as role models and advocates for physical activity. By creating supportive organizational, community, and policy environments, healthcare professionals can be better equipped to adopt active lifestyles, ultimately contributing to the reduction of non-communicable diseases and the promotion of healthier communities in Nigeria.

## Supporting information

**S1 File. Data and excerpts.**
(PDF)

## Acknowledgments

The authors express their sincere gratitude to the study participants for sharing their time and experiences. Appreciation is also extended to the field data collectors and research assistants for their dedication during data collection and to the Lagos University Teaching Hospital and the Lagos State Ministry of Health for their support and collaboration throughout the study.

## Author contributions

**Conceptualization:** Blossom Maduafokwa.

**Data curation:** Blossom Maduafokwa.

**Formal analysis:** Blossom Maduafokwa, Mobolanle Balogun, Kamaldeen Sunkanmi Abdulraheem.

**Funding acquisition:** Blossom Maduafokwa, Mobolanle Balogun, Kamaldeen Sunkanmi Abdulraheem.

**Investigation:** Blossom Maduafokwa, Mobolanle Balogun, Kamaldeen Sunkanmi Abdulraheem.

**Methodology:** Blossom Maduafokwa, Mobolanle Balogun, Kamaldeen Sunkanmi Abdulraheem.

**Project administration:** Blossom Maduafokwa, Mobolanle Balogun.

**Resources:** Blossom Maduafokwa, Mobolanle Balogun, Kamaldeen Sunkanmi Abdulraheem.

**Software:** Blossom Maduafokwa, Mobolanle Balogun, Kamaldeen Sunkanmi Abdulraheem.

**Supervision:** Blossom Maduafokwa, Mobolanle Balogun.

**Validation:** Blossom Maduafokwa, Mobolanle Balogun, Kamaldeen Sunkanmi Abdulraheem.

**Visualization:** Blossom Maduafokwa, Mobolanle Balogun, Kamaldeen Sunkanmi Abdulraheem.

**Writing – original draft:** Blossom Maduafokwa, Kamaldeen Sunkanmi Abdulraheem.

**Writing – review & editing:** Blossom Maduafokwa, Mobolanle Balogun, Kamaldeen Sunkanmi Abdulraheem.

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
