## [Decision Letter · Decision Letter 0]

16 Dec 2025

PONE-D-25-60319Facilitators and Barriers of Physical Activity, Sedentariness and Exercise Adoption Among Healthcare Professionals in Lagos State, Nigeria – A Qualitative ReviewPLOS One

Dear Dr. Maduafokwa,

Thank you for submitting your manuscript to PLOS ONE. After careful consideration, we feel that it has merit but does not fully meet PLOS ONE’s publication criteria as it currently stands. Therefore, we invite you to submit a revised version of the manuscript that addresses the points raised during the review process.

1. Please submit a fully or substantially completed COREQ or SRQR Checklist as this is a journal requirement for all Qualitative studies.2.Kindly submit copies of the two or all  Ethical approval letters from all Research Ethics Committees that approved this study.3. Please address all issues/comments raised by both peer reviewers.

We look forward to receiving your revised manuscript.

Kind regards,

Sylvester Chidi Chima, M.D., L.L.M, LLD

Academic Editor

PLOS One

Journal Requirements:

https://journals.plos.org/plosone/s/file?id=ba62/PLOSOne_formatting_sample_title_authors_affiliations.pdexf

“No competing interests are declared.”

3. In the online submission form, you indicated that [All relevant data supporting the findings of this study are contained within the manuscript. Additional anonymized transcripts and qualitative datasets may be made available upon reasonable request to the corresponding author (Dr. Blossom Adaeze Maduafokwa, bmaduafokwa@yahoo.com)].

Reviewers' comments:

Reviewer's Responses to Questions

**Comments to the Author**

1. Is the manuscript technically sound, and do the data support the conclusions?

Reviewer #1: Yes

Reviewer #2: Partly

2. Has the statistical analysis been performed appropriately and rigorously? 

Reviewer #1: Yes

Reviewer #2: Yes

3. Have the authors made all data underlying the findings in their manuscript fully available?

Reviewer #1: No

Reviewer #2: Yes

4. Is the manuscript presented in an intelligible fashion and written in standard English?

Reviewer #1: Yes

Reviewer #2: Yes

5. Review Comments to the Author

Reviewer #1: Comments to the Authors

Thank you for the opportunity to review this important and contextually relevant qualitative study on barriers and facilitators to physical activity among healthcare professionals in Lagos, Nigeria. The topic is timely and highly relevant for workforce health, productivity, and downstream patient counselling. The manuscript has strong potential, but several areas require clarification, methodological strengthening, and improved transparency before it can meet PLOS ONE publication standards.

1. Ethics approval inconsistencies

There are two different HREC approval numbers in the manuscript. Please correct the record and ensure the exact name of the committee, the single valid approval number, and the date of approval are clearly stated. Confirm that procedures described (audio-recording, consent process, data use) match the approved protocol.

2. Recruitment, sampling, and saturation need more detail

The manuscript states purposive sampling and that sample size was “guided by saturation,” but it lacks detail on:

• How participants were recruited (gatekeepers, public postings, direct invitations).

• Numbers invited vs. enrolled, and reasons for non-participation (if known).

• What constituted your saturation criterion and when it was reached.

Providing a brief recruitment flow diagram or paragraph would enhance transparency.

3. Strengthen description of analytic procedures

To meet reproducibility standards, please expand on:

• Who conducted coding, their training, and whether coding was double-checked.

• How discrepancies were resolved.

• Which software was used (e.g., NVivo, ATLAS.ti) or manual coding.

4. Data availability requirements

PLOS ONE requires underlying data to be publicly available unless restricted for ethical reasons. The current “available upon request” statement is insufficient.

Please either:

(a) deposit de-identified transcripts or coded datasets in a recognized repository (OSF, Figshare, institutional repository), providing DOI/accession numbers, or

(b) provide a justification for restricted access and describe the exact mechanism for data requests.

5. Reflexivity and researcher positionality

Please clarify:

• Who conducted the FGDs/IDIs/KIIs and their relationship to participants.

• Languages used, whether translators were involved, and how translation/back-translation quality was ensured.

• Whether member checking or participant validation was used.

This improves the manuscript’s trustworthiness.

6. Reporting guidelines

Please submit a completed COREQ checklist as supporting information to ensure full adherence to qualitative reporting standards.

7. Strengthen data–claim alignment

The findings are relevant and coherent, but several policy recommendations (e.g., onsite gyms, scheduled work breaks, intersectoral urban-planning solutions) should be tied more explicitly to participants’ quotes or existing evidence. Avoid generalizing beyond what qualitative data can assert.

8. Minor edits

• Ensure consistent abbreviations and formatting across tables.

• Add identifiers to all participant quotations for traceability.

• Light proofreading for grammar and clarity.

Overall assessment:

This is a meaningful and well-conceptualized qualitative study with clear public health value. With improvements to methodological transparency, data-management documentation, and alignment to qualitative reporting standards, the manuscript could be suitable for publication.

Reviewer #2: The study was well conducted. However there are some concerns with the methodolgy.

1. You need to state clearly how participants were recruited for the FGD, IDI, and KII.

2. You need to expand on the data collection process. How was the FGD, IDI, and KII conducted?

3. How and when was saturation achieved?

4. How many sessions of FGD, IDI and KII were held and why did you stop.

5. How was data analysed? You need to expand on the analysis of the data

6. PLOS authors have the option to publish the peer review history of their article (what does this mean? ). If published, this will include your full peer review and any attached files.

**Do you want your identity to be public for this peer review?** For information about this choice, including consent withdrawal, please see our Privacy Policy .

Reviewer #1: No

Reviewer #2: **Yes:** Tijani Oseni

---

## [Author Response · Author response to Decision Letter 1]

6 Jan 2026

Dear Editor,

We sincerely thank the Reviewers for their thorough evaluation of our manuscript and for the insightful, constructive comments provided. These comments have greatly improved the clarity, rigor, and overall quality of the paper. We have carefully considered each suggestion and have revised the manuscript accordingly. Below, we provide a point-by-point response, addressing all queries and indicating where corresponding changes have been made in the revised manuscript. All modifications are clearly marked, and we trust that the revised version satisfactorily meets the journal’s expectations. We remain grateful for the time and expertise invested in strengthening our work.

Sincerely,

Authors

Point to Point Response

1. Please submit a fully or substantially completed COREQ or SRQR Checklist as this is a journal requirement for all Qualitative studies.

Response: This has been done.

2. Kindly submit copies of the two or all Ethical approval letters from all Research Ethics Committees that approved this study.

Response: This has been done.

https://journals.plos.org/plosone/s/file?id=ba62/PLOSOne_formatting_sample_title_authors_affiliations.pdexf

Response: This has been done.

“No competing interests are declared.”

Response: This has been done.

5. In the online submission form, you indicated that [All relevant data supporting the findings of this study are contained within the manuscript. Additional anonymized transcripts and qualitative datasets may be made available upon reasonable request to the corresponding author (Dr. Blossom Adaeze Maduafokwa, bmaduafokwa@yahoo.com)].

Response: Data supporting the findings of this study are now presented in the supplementary file. The qualitative findings are based on focus group discussions, in-depth interviews, and key informant interviews conducted with healthcare professionals, their family members, healthcare leaders and policymakers.

Due to the sensitive nature of the qualitative data, which include detailed personal experiences, workplace practices, and professional roles, full public deposition of verbatim transcripts could risk deductive disclosure and compromise participant confidentiality, in accordance with the protocol approved by the research ethics committee.

To ensure transparency while maintaining ethical compliance, we have:

• Provided excerpts of anonymized participant quotations within the manuscript to support all themes and findings.

• Uploaded the interview guides, coding framework, and thematic structure as supplementary materials to allow reproducibility and methodological scrutiny.

Response: Thank you for this observation. The correct Ethical approval number HREC No: ADM/DSCST/HREC/APP/4115 has been indicated on page 12, line 44 only.

7. Have the authors made all data underlying the findings in their manuscript fully available?

Response: This has been addressed above.

8. Ethics approval inconsistencies

There are two different HREC approval numbers in the manuscript. Please correct the record and ensure the exact name of the committee, the single valid approval number, and the date of approval are clearly stated. Confirm that procedures described (audio-recording, consent process, data use) match the approved protocol.

Response: Thank you for this observation. The correct Ethical approval number HREC No: ADM/DSCST/HREC/APP/4115 has been indicated on page 12, line 44 only.

9. Recruitment, sampling, and saturation need more detail

The manuscript states purposive sampling and that sample size was “guided by saturation,” but it lacks detail on:

• How participants were recruited (gatekeepers, public postings, direct invitations).

• Numbers invited vs. enrolled, and reasons for non-participation (if known).

• What constituted your saturation criterion and when it was reached.

Providing a brief recruitment flow diagram or paragraph would enhance transparency.

Response: The manuscript now clearly describes how participants were recruited, including the use of facility gatekeepers, direct invitations, and referrals for healthcare professionals, family members, organizational leaders, and policymakers. We also added information on the number of individuals invited versus enrolled across study components, along with known reasons for non-participation, primarily related to scheduling and workload constraints. In addition, we clarified our saturation criterion, defining saturation as the point at which no new themes or substantive insights emerged from successive interviews or focus group discussions, and specified when saturation was reached for FGDs, IDIs, and KIIs. A brief recruitment flow paragraph has been included to enhance transparency. We appreciate the reviewer’s suggestion, which has strengthened the methodological rigor and clarity of the manuscript.

The recruitment methods have been outlined on page 2, lines 50 – 54 to page 3, lines 1 - 19

10. Strengthen description of analytic procedures

To meet reproducibility standards, please expand on:

• Who conducted coding, their training, and whether coding was double-checked.

• How discrepancies were resolved.

• Which software was used (e.g., NVivo, ATLAS.ti) or manual coding.

Response: We have expanded the Data Analysis section to clearly describe who conducted the coding, their training, and the procedures used to enhance analytic rigor. We now state that coding was primarily conducted by the lead researcher, who has formal training in qualitative research methods and thematic analysis, as well as professional certification in group fitness instruction and licensure as a Zumba instructor, providing relevant contextual familiarity with physical activity behaviors. To enhance analytic rigor and credibility, supervisors with expertise in qualitative research provided expert review of the coding framework, theme development, and interpretive coherence of the findings. The analytic process was strengthened through iterative expert review and peer debriefing. Supervisors examined the alignment between the coded data, emerging themes, and interpretations, and provided structured feedback. Any discrepancies or areas of concern identified during this review were discussed and resolved through analytic refinement, including clarification of code definitions, consolidation or separation of themes where necessary, and re-examination of supporting data extracts.

Page 5, Lines 28 – 32

11. Data availability requirements

PLOS ONE requires underlying data to be publicly available unless restricted for ethical reasons. The current “available upon request” statement is insufficient.

Please either:

(a) deposit de-identified transcripts or coded datasets in a recognized repository (OSF, Figshare, institutional repository), providing DOI/accession numbers, or

(b) provide a justification for restricted access and describe the exact mechanism for data requests.

Response: This has been addressed above.

12. Reflexivity and researcher positionality

Please clarify:

• Who conducted the FGDs/IDIs/KIIs and their relationship to participants.

• Languages used, whether translators were involved, and how translation/back-translation quality was ensured.

• Whether member checking or participant validation was used.

This improves the manuscript’s trustworthiness.

Response: We have revised the manuscript to provide a clearer and more comprehensive description of reflexivity and researcher positionality in line with COREQ guidelines. All focus group discussions (FGDs), in-depth interviews (IDIs), and key informant interviews (KIIs) were conducted by the lead researcher, a public health physician with formal training in qualitative research methods and thematic analysis. The researcher had no prior personal or professional relationships with participants, and no authority or supervisory roles existed between the researcher and participants at the time of data collection. This has now been explicitly stated in the manuscript. We have also clarified that data collection was conducted solely in English, and no interpreters were used.

Regarding participant validation, formal member checking was not conducted due to logistical constraints. This has been transparently acknowledged in the manuscript. We have detailed alternative strategies used to ensure rigor, including iterative reflexive analysis, peer debriefing, use of verbatim participant quotations to support themes, and maintenance of a detailed audit trail documenting analytic decisions.

Page 3, lines 53 - 56

13. Reporting guidelines

Please submit a completed COREQ checklist as supporting information to ensure full adherence to qualitative reporting standards.

Response: This has been done.

14. Strengthen data–claim alignment

The findings are relevant and coherent, but several policy recommendations (e.g., onsite gyms, scheduled work breaks, intersectoral urban-planning solutions) should be tied more explicitly to participants’ quotes or existing evidence. Avoid generalizing beyond what qualitative data can assert.

Response: We have removed any over-generalisations and revised the manuscript to ensure that policy recommendations are clearly framed as participant-derived suggestions rather than definitive or universal prescriptions. Recommendations such as on-site gyms, scheduled work breaks, and intersectoral urban-planning approaches were explicitly raised by participants during the FGDs and KIIs, and this is now made clearer in the text.

To maintain brevity, not all illustrative quotations were included; however, the recommendations presented are grounded in the qualitative data and aligned with the identified themes. We have also adjusted the language throughout to avoid extending interpretations beyond what the qualitative evidence can support.

15. Minor edits

• Ensure consistent abbreviations and formatting across tables.

• Add identifiers to all participant quotations for traceability.

• Light proofreading for grammar and clarity.

Response: We have ensured consistent use of abbreviations and formatting across all tables and added identifiers to all participant quotations to enhance traceability. Regarding language, participant quotations have been intentionally left verbatim and not grammatically corrected, in keeping with qualitative research best practice, to preserve the authenticity and meaning of participants’ voices. Minor grammatical and stylistic edits were applied only to the authors’ narrative text to improve overall clarity and readability.

1. You need to state clearly how participants were recruited for the FGD, IDI, and KII.

Response: This has been addressed above. Page 2, Lines 50 – 54 to Page 3, Lines 1 - 4

2. You need to expand on the data collection process. How was the FGD, IDI, and KII conducted?

Response: We have expanded the Data Collection section to provide a more detailed description of how the FGDs, IDIs, and KIIs were conducted. Specifically, we have clarified the rationale for each method and development of each tool based on the Social Ecological Model, participant composition and setting for FGDs, procedures for moderating discussions and managing group dynamics, and the conduct of IDIs and KIIs, including selection of participants, interview settings, duration, and use of probes. We have also detailed the language of data collection, audio-recording procedures, use of field notes, informed consent, and the process for determining thematic saturation.

Page 3, Lines 30 – 51

3. How and when was saturation achieved?

Response: We have clarified the process for determining data saturation in the manuscript. Saturation was assessed concurrently with data collection and analysis through ongoing review of transcripts and field notes. After successive FGDs, IDIs, and KIIs, no new codes or substantive themes emerged, and existing themes were consistently reinforced across participant groups. Recruitment and data collection were discontinued at the point where additional interviews did not yield new insights, indicating that thematic saturation had been achieved. This clarification has been added to the Data Collection section of the manuscript.

Page 3, Lines 50 – 53

4. How many sessions of FGD, IDI and KII were held and why did you stop.

Response: We have clarified the number of FGDs, IDIs, and KIIs conducted in the revised manuscript. In total, 6 FGDs, 4 IDIs, and 5 KIIs were conducted. Data collection proceeded alongside concurrent analysis, and recruitment was discontinued once thematic saturation was achieved. Specifically, successive sessions no longer yielded new codes or substantive insights across the levels of the Social Ecological Model, indicating that additional data collection was unlikely to contribute further analytical depth. This information has now been added to the Data Collection section of the manuscript.

Page 3, Lines 4 - 18

5. How was data analysed? You need to expand on the analysis of the data

Response: The Data Analysis section has been expanded to provide a more detailed description of the analytic process. Specifically, we have clarified that data were analysed using a thematic analysis approach guided by the Social Ecological Model, incorporating both deductive and inductive coding. We have described the development of the initial coding framework, the iterative process of coding and theme refinement, and the use of both software-assisted (Dovetail) and manual analytic techniques, including memo writing, code mapping, and theme clustering. We have also detailed the involvement of multiple coders. These revisions have been incorporated into the Data Analysis section of the manuscript.

Page 3, Lines 5 - 25

---

## [Decision Letter · Decision Letter 1]

19 Jan 2026

Facilitators and Barriers of Physical Activity, Sedentariness and Exercise Adoption Among Healthcare Professionals in Lagos State, Nigeria – A Qualitative Review

PONE-D-25-60319R1

Dear Dr. Maduafokwa,

We’re pleased to inform you that your manuscript has been judged scientifically suitable for publication and will be formally accepted for publication once it meets all outstanding technical requirements.

Kind regards,

Sylvester Chidi Chima, M.D., L.L.M, LLD

Academic Editor

PLOS One

Reviewers' comments:

Reviewer's Responses to Questions

**Comments to the Author**

1. If the authors have adequately addressed your comments raised in a previous round of review and you feel that this manuscript is now acceptable for publication, you may indicate that here to bypass the “Comments to the Author” section, enter your conflict of interest statement in the “Confidential to Editor” section, and submit your "Accept" recommendation.

Reviewer #1: All comments have been addressed

Reviewer #2: All comments have been addressed

2. Is the manuscript technically sound, and do the data support the conclusions?

Reviewer #1: Yes

Reviewer #2: Yes

3. Has the statistical analysis been performed appropriately and rigorously? 

Reviewer #1: Yes

Reviewer #2: Yes

4. Have the authors made all data underlying the findings in their manuscript fully available?

Reviewer #1: Yes

Reviewer #2: Yes

5. Is the manuscript presented in an intelligible fashion and written in standard English?

Reviewer #1: No

Reviewer #2: Yes

6. Review Comments to the Author

Reviewer #1: All comments have been duly addressed in the revised manuscript. Authors should ensure that the proofs are read meticulously to avoid errors in the final manuscript.

Reviewer #2: (No Response)

7. PLOS authors have the option to publish the peer review history of their article (what does this mean? ). If published, this will include your full peer review and any attached files.

**Do you want your identity to be public for this peer review?** For information about this choice, including consent withdrawal, please see our Privacy Policy .

Reviewer #1: **Yes:** Omoladun Olukemi Odediran

Reviewer #2: **Yes:** Tijani Idris Ahmad Oseni

---

## [Editor Report · Acceptance letter]

PONE-D-25-60319R1

PLOS One

Dear Dr. Maduafokwa,

I'm pleased to inform you that your manuscript has been deemed suitable for publication in PLOS One. Congratulations! Your manuscript is now being handed over to our production team.

Kind regards,

on behalf of

Professor Sylvester Chidi Chima

Academic Editor

PLOS One